# Cellulose Nanocrystal Surface Cationization: A New Fungicide with High Activity against *Phycomycetes capsici*

**DOI:** 10.3390/molecules24132467

**Published:** 2019-07-04

**Authors:** Shunyu Xiang, Xiaozhou Ma, Shuyue Liao, Huan Shi, Changyun Liu, Yang Shen, Xing Lv, Mengting Yuan, Guangjin Fan, Jin Huang, Xianchao Sun

**Affiliations:** 1College of Plant Protection, Southwest University, Chongqing 400715, China; 2Chongqing Key Laboratory of Soft-Matter Material Chemistry and Function Manufacturing, Southwest University, Chongqing 400715, China; 3School of Chemistry and Chemical Engineering, Southwest University, Chongqing 400715, China

**Keywords:** cellulose nanocrystal, hexadecyl trimethyl ammonium bromide, quaternary ammonium salt, nano-antifungal material, *Phytophthora capsici*

## Abstract

At present, the management of *Phytophthora capsici* (*P. capsici*) mainly relies on chemical pesticides. However, along with the resistance generated by *P. capsici* to these chemical pesticides, the toxicity and non-degradability of this chemical molecule may also cause serious environmental problems. Herein, a new bio-based nano-antifungal material (CNC@CTAB) was made with coating hexadecyl trimethyl ammonium bromide (CTAB) on the surface of a cellulose nanocrystal (CNC). This material was then applied to the prevention of *P. capcisi*. This particle was facilely fabricated by mixing CTAB and sulfuric group modified CNC in an aqueous solvent. Compared to pure CTAB, the enrichment of CTAB on the CNC surface showed a better anti-oomycete activity both in vitro and in vivo. When CNC@CTAB was applied on *P. capsici* in vitro, the inhibition rate reached as high as 100%, while on the pepper leaf, the particle could also efficiently prevent the infection of *P. capsici*, and achieve a disease index as low as zero Thus, considering the high safety of CNC@CTAB in agricultural applications, and its high anti-oomycete activity against *P. capsici*, we believe that this CNC@CTAB has great application potential as a new green nano-fungicide in *P. capsici* management during the production of peppers or other vegetables.

## 1. Introduction

*Phytophthora capsici* (*P. capsici*), an attractive model oomycete plant pathogen with a widely broad host range, attacks solanaceous (peppers, tomatoes), legume (*lima* and *snap beans*), and most cucurbit hosts, which results in a great annual loss to agriculture production [1,2,3]. *P. capsici* can be dispersed within the soil and surface water and survive in the field for several years [4]. Moreover, this pathogen has high variation with its sexual life cycle, making disease management and prevention difficult [5]. Many broad-spectrum fungicides are not effective against this oomycete for their special physiological and biochemical structures [6,7]. Until now, the prevention of the *P. capsici* mainly relied on chemical pesticides, with few alternatives available [7,8]. Recently, some chemicals specifically targeting *P. capsici* and other oomycetes (e.g., etridiazole, metalaxyl, mefenoxam, and fosetyl-Al) been introduced to improve the management of *Phytophthora* diseases [8]. However, the rapid regeneration times and exceptional adaptability of *P. capsici* have caused the development of fungicide resistance within specific pathogen populations [9,10,11,12,13,14]. Meanwhile, the application of these chemical fungicides is causing serious environmental pollution because of their toxicity, low degradability, and the high fungicide tolerance of oomycetes [5]. Hence, the discovery of green, efficient, and novel fungicides to manage oomycetes is an important priority in fungicide research.

Hexadecyl trimethyl ammonium bromide (CTAB), an efficiency anti-microbial surfactant, has been deeply studied and applied for several decades [15,16,17]. Its antimicrobial mechanisms can be interpreted as follows: the quaternary ammonium head of CTAB enables the molecule to quickly adsorb onto the cell surface. Thus, its hydrophobic moieties interact with the phospholipid bilayer of the cell membrane and destroy the self-assembled structure of the pathogeny cell membrane, leading to a disruption of the cell membrane and a release of cell constituents, which eventually leads to cell death [18,19]. CTAB has a relatively low toxicity and is hard to be resisted by microorganisms [17,20]. This characteristic allows it to be widely used as a nonspecific anti-microbial agent for water disinfection, bioactive agents, medical apparatuses and instruments, antifouling coatings, and other antimicrobial works [17,18,21,22,23,24]. However, it also been applied on oomycetes and been proven to have an anti-oomycete activity [25]. However, as an oomycete inhibitor, its activity is highly dependent on its concentration [25]. Therefore, as a kind of pesticides, the high application amount of CTAB during plant disease control is still an important problem that remains to be solved.

Rod-like nanoparticles have been shown to have a strong shape-dependent interaction with cell membranes and can be used in targeted intracellular drug delivery [26,27]. Cellulose nanocrystals (CNCs), as a type of biomass-based rod-like nanoparticle, have also shown some advantages, as they have high biocompatibility, low toxicity, and biodegradability in comparison with inorganic nanoparticles. CNCs usually show a high aspect ratio, with a length of 100–2000 nm and a diameter of 20–40 nm [28]. Considering that the interaction ability of the rod-like nanoparticles with the cell membrane is closely related to its aspect ratio (i.e., a high aspect ratio will usually lead to stronger interactions), CNCs show great penetration, and, thus, might act as ideal potential biological carriers [29,30,31,32], for example, in the application of cosmetic drugs [32,33,34,35]. On the other hand, the rich positive charge and hydroxyl groups on a CNC’s surface also mean the particle could be modified by Quaternary ammonium salt (e.g., CTAB) and applied in high-performance composite polymers [36,37]. The composite nanoparticles prepared by CTAB and CNC are proven to be able to reverse the zeta potential of CNCs from negative to positive, and thus increase the monodisperse rate of the particle, and also adjust the surface hydrophilicity of the particle. These results can be usefully applied in the adjustment of the compatibility between CNCs and polymer matrices to prepare high-performance CNC-based composite materials [36,37]. For example, CTAB can be used to modify the surface of CNCs by electrostatic interaction. After modification, the zeta potential of the CNC is changed from -36.5mV to +6.09mV. Meanwhile, the hydrophobicity of the CNC also increases with the improvement of its dispersibility in the polypropylene (PP) matrix. the composite CNC can be used to prepare a high-performance PP, and the Young’s modulus of the PP is reported to increase from 181 MPa to 279 MPa [38]. However, the anti-microbial activity of the CTAB coated CNC (CNC@CTAB) has not been studied. This activity might be very useful in the prevention of plant disease. Meanwhile, CTAB can also act as stabilizer of both CNC and inorganic nanoparticles [39,40], for example, a CNC/silver nanoparticle/CTAB composite nanoparticle, and is used in PLA film to be applied in anti-bacterial functions. CTAB, in this work, highly improved the stability of silver nanoparticle (SNP) and CNC, while the composite film showed good anti-bacterial activity [39]. However, although SNP has been proven to exhibit high anti-bacterial activity, the ability of CTAB in this material to inhibit microbial development is not well studied.

By coating CTAB onto the surface of CNCs, the local concentration of CTAB can be increased on the CNC’s surface, while the interaction between cell membrane and the rod-like CNC@CTAB particle can also increase the anti-microbial activity of the CTAB. Thus, in this work, the anti-oomycete activity of CNC@CTAB was studied. We choose *P. capsici* as the model oomycete. The surface density of CTAB on the CNC@CTAB particle was adjusted and calculated by the N content of the CNC@CTAB particle. Then, the inhibition of pure CTAB and CNC@CTAB with a different CTAB surface density to *P. capsica*, both in vitro and in vivo, was compared. Meanwhile, we also studied the conductivity change of the *P. capsici* cell membrane treated with CTAB and CNC@CTAB, respectively. Our results showed that after coating CTAB onto CNCs, the inhibition activity of the particle was obviously higher than that of the pure CTAB, and closely depended on the surface CTAB density.

## 2. Results and Discussion

### 2.1. Characterization of CNC and CNC@CTAB

To achieve CNCs with high surface negative charges, the CNC was extracted via sulfuric acid hydrolysis form cotton fiber, which could modify sulfuric groups onto the CNC’s surface. Then, CTAB with a different concentration was mixed with CNCs under pH 10 to prepare CNC@CTAB with a different CTAB surface density. Fourier transform infrared spectroscopy (FTIR) spectroscopy was used to observe CTAB molecules on the particle surface [24]. As is shown in Figure 1a, after mixing CTAB with CNC, the peaks located at 2900 cm^−1^ and 2860 cm^−1^ corresponded to the symmetric and asymmetric C-H stretching vibrations, which were slightly strengthened. Compared with the FTIR spectrum of CTAB, this feature was attributed to the formation of the CTAB layer on the surface of the CNC. Meanwhile, the interaction of the CTAB and the CNC led to a decrease in the strength of the characteristic peak to 1645 cm^−1^, corresponding to the decreased water content in the samples. This result was due to the greater hydrophobic behavior of CNC@CTAB. This result was also inconsistent with the formal reports and proved that the CTAB was successfully coated on the CNC surface [41]. However, as there was no chemical reaction happened between CNC and CTAB, the coating of CTAB onto the CNC surface showed no effect on the crystalline structure of the CNC (Figure 1b). The result of X-ray diffraction (XRD) was also consistent with other related reports [38]. Based on the transmission electron microscope (TEM) images of the CNC and the CNC@CTAB (Figure 1c,d) it could also be observed that, after coating, the length of the particle was not obviously changed, while the width of the particle was increased, which was due to the surface CTAB layer of the CNC@CTAB. On the other hand, the coating of the CTAB slightly aggregated the particle, which may be due to the change in the surface hydrophilicity of the particle induced by the CTAB layer [36]. Meanwhile, atomic force microscope (AFM) was used to analyze the width and length of the particle. The CNC was found to be 180.1 ± 7.6 nm long and 21.6 ± 1.8 nm wide. After coating with CTAB, the length of the CNC@CTAB did not change (173.6 ± 6.1 nm); however, the diameter of the CNC@CTAB was increased to 32.5 ± 1.6 nm, mainly owing to the existence of the CTAB layer on the nanoparticle.

The coating of CTAB or quaternary ammonium salt onto the CNC surface also affected the zeta potential of the particle [38,42,43,44]. As is shown in Table 1, with an increase of the added CTAB amount, the zeta potential of the CNC@CTAB gradually changed from negative (−39.4 mV) to positive (+7.7 mV). This was mainly due to the increase in the positive charge amount of CTAB on the CNC surface. A positive surface charge can make the particle more easily interact with the cell surface and thus help the CTAB to destroy the cell membrane of the pathogen. The N% content of CNC@CTAB was used to calculate the surface density of the CTAB. Table 1 listed the N% content of the CNC and CNC@CTAB with different surface densities. It was found that with an increase in the added CTAB amount, the N% content of the particle increased, which indicated that the content of the CTAB in the CNC@CTAB increased. After calculation, by controlling the added amount of CTAB during the mixing process, the CTAB density on the CNC surface could be controlled from 2.86 × 10^−2^ nmol/cm^2^ to 5.50 × 10^−2^ nmol/cm^2^. The dispersibility of the particle, at the same time, was not affected by the CTAB coating. As is shown in Figure 1e, after being coated with CTAB, the CNC@CTAB could still be dispersed well in water, while the particles could be suspended in water evenly after being dispersed for 2h. This occurs because, in the CTAB layer, the ammonium head of the CTAB molecules moved outside of the particle, which not only gave the particle enough hydrophilicity, but also provided the particle a high surface charge that could ensure that the CNC@CTAB stably disperses in the aqueous solvent [37].

### 2.2. The Activity of CNC, CTAB, and CNC@CTAB against Phytophthora capsici

Theoretically, the enrichment of the CNC with the CTAB on its surface might improve the anti-oomycete activity of the CTAB. However, CNC itself was reported to have no effect on the cells. Thus, the anti-oomycete activity of the CNC was tested (Appendix A), and then the inhibition of CNC@CTAB and pure CTAB on *P. capsici* was compared (Figure 2). This result showed that pure CNC would not inhibit the hyphae growth of *P. capsici*, even at a high concentration of 2 mg/mL. However, both pure CTAB and CNC@CTAB had an obvious inhibitory effect on *P. capsici*. As is shown in Figure 2a, with an increase in pure CTAB and CNC@CTAB concentrations, the inhibitory effect against *P. capsici* was also increased. Meanwhile, it could also be found that the inhibitory effect of CNC@CTAB was much higher than pure CTAB when the CTAB concentration of the two experimental groups was same; this can also be found in Figure 2b. As seen in Figure 2b, the inhibitory effect of CNC@CTAB against *P. capsici* was almost two times of that of pure CTAB when the CTAB concentration of the two groups was same. On the other hand, the surface density of the CTAB on the CNC surface can also affect the anti-oomycete activity of the CNC@CTAB. With an increase in CTAB surface density, the anti-oomycete activity of CNC@CTAB was also increased. When the surface density of CTAB increased from 2.86 × 10^−2^ nmol/cm^2^ to 5.50 × 10^−2^ nmol/cm^2^, the inhibition rate of CNC@CTAB increased from 69.1% to 100% when the total CTAB concentration of the CNC@CTAB particles was 74 μg/mL. This is because increasing the CTAB surface density not only reverses the zeta potential of the CNC@CTAB, and makes the particle much easier to interact with the cell surface of *P. capsici*, but also forms a pre-self-assembled CTAB layer on the CNC surface, which could efficiently be inserted into the cell membrane and destroy its structure.

### 2.3. CNC@CTAB Damage Fungal Cell Membrane Permeability

Figure 3 shows the permeability change of the *P. capsici* cell membrane after CNC, pure CTAB, and CNC@CTAB were added with different CTAB surface densities. After the CTAB and particles were added for 24 h, it was seen that, compared to the control group (CK), the CNC showed no effect on the cell membrane’s permeability. However, both pure CTAB and CNC@CTAB obviously increased the permeability of the cell membrane by damaging the membrane’s structure. The change of the cell membrane permeability of pure CTAB and CNC@CTAB groups was time dependent. However, interestingly, unlike the permeability change of the pure CTAB group, it was observed that the permeability of the CNC@CTAB groups showed a two-stage change. For the pure CTAB group, the cell membrane permeability change rate was stable, since the adsorption and assembling of CTAB on the cell membrane, as well as its destruction, happened simultaneously. On the contrary, it could be seen that after CNC@CTAB was added to *P. capsici*, cell membrane permeability was increased quickly in the first 6 h, and the increase rate was decreased to a lower level. This indicated that the particle could be quickly absorbed onto the cell surface and destroy the cell membrane. As CTAB was pre-assembled and enriched on the particle surface, the CNC@CTAB particle could directly damage the cell membrane, and the particle concentration, not the CTAB concentration, highly affected the damage rate. Thus, when CNC@CTAB concentration was at a high level, the particles quickly destroyed the cell membrane, and performed a high cell membrane permeability change rate, while when the free particle concentration gradually decreased due to the absorbance of particles onto the cell surface (and the cell membrane structure was destroyed to a very high level), the increase rate of the permeability decreased. Furthermore, with the increase of surface CTAB density, the permeability change rate of the cell membrane also increased. This was because the higher CTAB density gave the particle more surface positive charges, and the particle could more easily absorb onto the cell surface. Also, a thicker CTAB layer would make the particle easier to insert into the cell membrane and, thus, damage the self-assembling structure of the membrane. This result was also consistent with the inhibition activity of the CNC@CTAB particle against *P. capsici*. 

### 2.4. Inhibition of Pathogens Infecting Pepper Leaves

*P. capsici* can infect pepper roots, stems, leaves, and fruits. It mainly causes stem blight and wilt, as well as fruit and leaf necrosis [2]. Figure 4 compares the prevention of *P. capsici* by water (the CK group), CNC, CTAB, and CNC@CTAB with different surface CTAB densities, while the disease indexes of every group are given in Table 2. It was found that, in the CK, CNC, and CTAB groups, after being inoculated with *P. capsici*, the pepper leaves showed obvious necrosis. However, in the CNC@CTAB groups, the necrosis degree of the leaves was greatly decreased and could not even be observed. More specifically, for the CK and CNC groups, the disease indexes were as high as 16.7, while almost half of the leaves were infected. Meanwhile, CTAB showed some anti-oomycete activity. The necrosis area and degree decreased, and the disease index also decreased to 13. On the other hand, unlike the CK, CNC, and CTAB groups, with an increase in surface CTAB density, the necrosis degree of the leaves in the CNC@CTAB groups greatly decreased from 13 in the CTAB group to 1.5 (CNC@CTAB (1) group) or even 0 (CNC@CTAB (2) group and CNC@CTAB (3) group) (Table 2). This result indicates that the enrichment of CTAB on the CNC surface can rapidly inhibit the *P. capsica* infection of pepper leaves, and the disease prevention and control ability of the CNC@CTAB increased by increasing the density of the CTAB on the CNC surface.

## 3. Conclusions

The use of CNC as a nanocarrier to prevent plant disease is a novel methodology in the field of plant protection and agriculture. In the present study, by adjusting the surface CTAB density, the anti-oomycete activity of CNC@CTAB was studied. CTAB was simply coated onto CNC by mixing a sulfuric group modified CNC with CTAB. This strategy successfully turned CTAB, a small molecule anti-oomycete agent, into a bio-based nano-anti-oomycete material (CNC@CTAB). the CNC@CTAB particle then showed high anti-oomycete activity against *P. capsici*, while the surface CTAB density of the particle also closely affected the anti-oomycete activity of the CNC@CTAB. This high anti-oomycete activity was believed to be due to the enrichment of CTAB on the CNC surface and the nature of rod-like particles (like CNC and CNC@CTAB) to interact with cell surfaces. Meanwhile, the CTAB layer also provided the particle a large number of positive charges, which also helped the CNC@CTAB to interact with the cell surface. This method was proven by observing the change in cell membrane conductivity after adding CNC@CTAB, which corresponded to the cell membrane permeability and damage degree. The addition of the CNC@CTAB particle produced a two-stage cell membrane permeability change. The change rate was high during the first 6 h and then decreased to a moderate level. This indicated that the particle could be absorbed onto the cell surface and quickly damage the cell membrane. Further observations and studies were made on the in vivo anti-oomycete activity of CNC@CTAB on a pepper leaf. We compared the ability of water, CNC, pure CTAB and CNC@CTAB with different surface CTAB densities to prevent *P. capsici* infection on a pepper leaf. The data showed that the water group, CNC group, and CTAB group showed low or no activity in the prevention of *P. capsici* infection, while the CNC@CTAB with different surface CTAB densities showed excellent in vivo anti-oomycete activity. The anti-oomycete of the CNC@CTAB closely depended on its surface CTAB density, and the particle could even completely prevent the infection of *P. capcisi* on a pepper leaf (disease index = 0) when the surface CTAB density was higher than 3.96 × 10^−2^ nmol/cm^2^. Considering the high anti-oomycete activity and high safety of CNC and CTAB in agriculture applications, as well as its high anti-oomycete activity against *P. capsici*, we believe that this CNC@CTAB has great application potential as a new green nano-fungicide for *P. capsici* management during the production of peppers or other vegetables. 

## 4. Materials and Methods

### 4.1. Experimental Materials and Instruments 

Sulfuric acid (98%) was purchased from Chongqing Chuandong Chemical Co., Ltd. (Chongqing, China); sodium hydroxide was purchased from Chengdu Kelon Chemical Reagent Factory, Japan; Hexadecyl trimethyl ammonium bromide (purity > 99%) was purchased from Dingguo Changsheng Biotechnology Co., Ltd. (Beijing, China); and V8 vegetable juice (100%) was purchased from Campbell’s Soup Company (State of New Jersey, USA). Agar was purchased from Beijing Dingguo Changsheng Biotechnology Co., Ltd. (Beijing, China). Instrument information: a transmission electron microscope (TEM, JEM-1200EX; JEOL, Tokyo, Japan), Fourier infrared (Nicolet 6700, Madison, WI, USA, 400–4000 cm^−1^), and Zetasizer Nano ZS90 (Malvern Panalytical, Heracles Almelo, Netherlands), elemental analyzer (Vario EL cube, Elementar Analysensysteme, Frankfurt, Germany) were used in the present study.

### 4.2. Sulfuric Acid Extraction of Cellulose Nanocrystal (CNC)

A total of 40 g NaOH was weighed and dissolved in 2000 mL water to make a 2% NaOH solution. Then, 50 g cotton fiber was added to the NaOH solution and stirred at 25 °C for 12 h. The cotton treated with the NaOH solution was washed with water until neutral (measured with a pH meter) and then dried. A total of 126 mL of 90% concentrated sulfuric acid was added to 124 mL of distilled water under cool to prepare a 65% sulfuric acid solution. After the temperature was decreased to room temperature (25 °C), 12.5 g of the dried alkali-treated cotton was placed into the sulfuric acid solution and stirred at 45 °C for 1 h. The cotton was centrifuged for 7 days and then lyophilized for use.

### 4.3. CNC Grafted Cetyltrimethylammonium Bromide (CTAB) and Its Characterization

The CNC@CTAB was synthesized according to the methodology described by Malladi Nagalakshmaiah [38]. A total of 1 g CNC was weighed and dissolved in 100 mL water and sonicated for 15 min, and, then, the pH of the CNC solution was adjusted to 10 with 0.5 M NaOH. Samples of 25 mg, 50 mg, or 100 mg CTAB were added to the CNC solution and stirred at 40 °C for 3 h to produce CNC@CTAB with different graft ratios. Finally, the CNC@CTAB solution was centrifuged for 5 days. The CNC, CTAB, and CNC@CTAB were characterized by FTIR spectroscopy, elemental analysis, and zeta potential. The CNC and CNC@CTAB were characterized by Atomic force microscope (AFM), Transmission electron microscopy (TEM), and X-ray diffraction (XRD). The following formula (1) was used to calculate the density of the CTAB on the CNC surface:(1)$$\mathrm{Density}\;(\mathrm{nmol}/{\mathrm{cm}}^{2})\;=\;\frac{M}{S},$$
where M is the molar mass of CTAB on the surface of CNC@CTAB and S (250 m^2^/g) is the surface area of the CNC [45].

### 4.4. CNC@CTAB against Phytophthora capsici Activity

The CTAB content of the surface of the CNC@CTAB was calculated based on the elemental analysis results of the CNC@CTAB. Then, the CNC@CTAB and CTAB were added to the V8 medium to form a medium with the same concentration of CTAB (8.6 µg/mL, 17.2 µg/mL, 34.4 µg/mL, 68.8 µg/mL, and 74 µg/mL). The CNC and water groups were used as controls. *P. capsica* was incubated in the CNC@CTAB-containing or CTAB-containing medium with different concentrations for 10 days at 21 °C in an incubator. The following formula (2) was used to calculate the inhibition rate: (2)$$\mathrm{Inhibition}\;\mathrm{rate}\;(\%)\;=\;\frac{R-r}{R}\;\times \;100\%,$$
where R is the diameter of the clear water control group, and r is the diameter of the treatment group.

### 4.5. Electrical Conductivity Assay of Cell Membrane

The electrical conductivity of the mycelia was measured according to the methodology described by Firoz et al. [46]. *P. capsici* was cultivated in a V8 medium for 6 days. Then, ten mycelial plugs with diameters of 10 mm from fresh edges of six-day-old colonies were transferred into 100 mL V8 liquid medium (V8 medium without agar) and incubated at 21 °C and 180 r/min for 48 h. After 48 hours of cultivation, the CNC@CTAB (with different CTAB density) and CTAB were added into the medium (the total concentrations of CTAB in the CTAB and CNC@CTAB groups were the same) to make the concentration of CTAB in each group reach 74 μg/mL; pure CNC and water groups were the control groups. The electrical conductivity of each group was measured by an electrical conductivity meter at 0 h, 2 h, 4 h, 8 h, and 24 h. Each group was given five replicates. 

### 4.6. Experiment of Pepper Leaf Infection In Vitro

Peppers with consistent growth rates were selected. CNC@CTAB and CTAB with the same concentration of CTAB were sprayed on the surface of pepper leaves (50 mL per plant). After spraying for 24 h, the pepper leaves of the same size were selected and inoculated with *P. capsici* on the same part of the leaves and incubated at 21 °C for 72 h. Formula (3) was used to calculate the disease index. The grading criteria for the incidence of pepper leaves are according to Table 3.
(3)$$\mathrm{Disease}\;\mathrm{index}=\frac{\sum \left(N_{1}\;\times \;G\right)}{N\;\times {\;G}_{\max}}\times \;100\%$$
where N_1_ is the leaf counts at all levels, G is the number of leaves at this level, N is the total number of leaves, and G_max_ is the highest-grade representative value. 

## Figures and Tables

**Figure 1 molecules-24-02467-f001:**
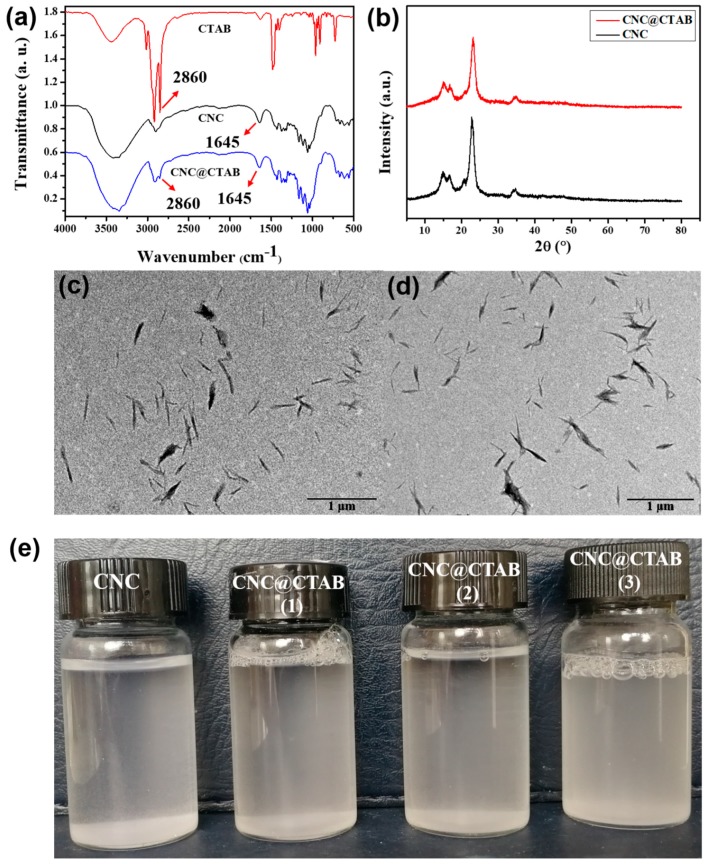
The Fourier transform infrared (FTIR) spectroscopy of the cellulose nanocrystal (CNC) and CNC@ coating hexadecyl trimethyl ammonium bromide (CTAB) (**a**). The X-ray diffraction patterns obtained for the CNC and CNC@CTAB (**b**). The transmission electron microscope (TEM) images for the CNC and CNC@CTAB (**c**,**d**). State of suspension of the CNC (0.1 wt%) and CNC@CTAB (0.1 wt%) in water for 2 h (**e**). CNC is the cellulose nanocrystal. CNC@CTAB is a material formed by the CTAB and CNC.

**Figure 2 molecules-24-02467-f002:**
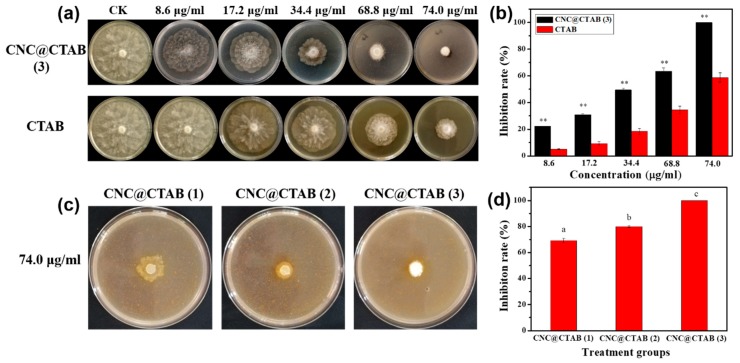
Comparison of the anti-oomycete activity of the CNC@CTAB and CTAB against *Phytophthora capsici* (**a**). The inhibition rate statistics of the CTAB and CNC@CTAB against *Phytophthora capsici* (**b**). Comparison of the anti-oomycete activity of the CNC@CTAB with different CTAB densities against *Phytophthora capsici* (**c**). The inhibition rate statistics of CNC@CTAB with different CTAB densities against *Phytophthora capsici* (**d**). CNC@CTAB (3) means that the CTAB density on the CNC surface is 5.50 × 10^−2^ nmol/cm^2^. CNC@CTAB (2) means that the CTAB density on the CNC surface is 3.96 × 10^−2^ nmol/cm^2^. CNC@CTAB (1) means that the CTAB density on the CNC surface is 2.86 × 10^−2^ nmol/cm^2^. CTAB is the hexadecyltrimethylammonium bromide. Mean values displayed in each bar followed by different letters are significantly different according to Duncan’s multiple range test (*p* < 0.05). ** indicates separation among the total CTAB concentration of the CNC@CTAB and pure CTAB at the same concentration by Duncan multiple comparison (**: *p* < 0.01). vertical bars indicate standard deviations (*n* = 3) ± S.E.

**Figure 3 molecules-24-02467-f003:**
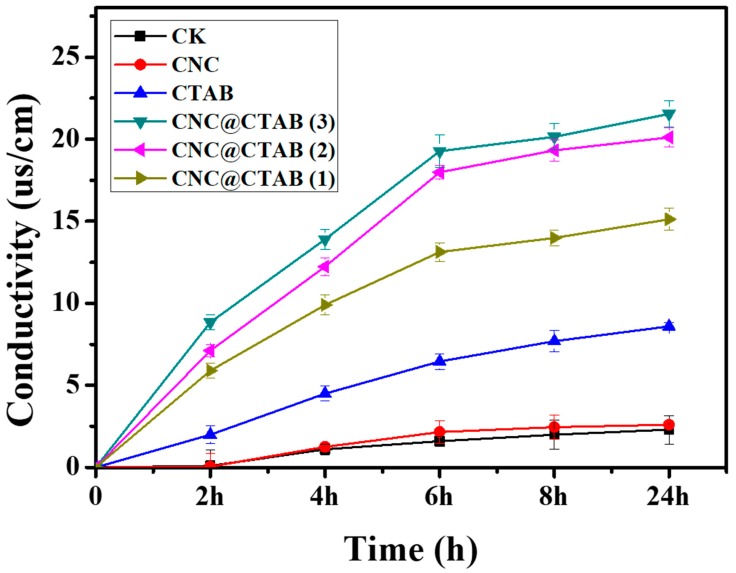
Curve of the extracellular fluid conductivity of *Phytophthora capsici* with time after adding CNC, CTAB, and CNC@CTAB (with different CTAB densities). CNC@CTAB (3) means that the CTAB density on the CNC surface is 5.50 × 10^−2^ nmol/cm^2^. CNC@CTAB (2) means that the CTAB density on the CNC surface is 3.96 × 10^−2^ nmol/cm^2^. CNC@CTAB (1) means that the CTAB density on the CNC surface is 2.86 × 10^−2^ nmol/cm^2^. The CNC is the cellulose nanocrystal. CTAB is the hexadecyltrimethylammonium bromide. Control group (CK) is the water group. Vertical bars indicate standard deviations (*n* = 3) ± S.E.

**Figure 4 molecules-24-02467-f004:**
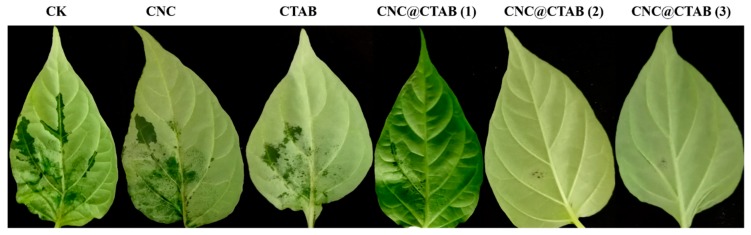
In vitro test of pepper leaves. CNC@CTAB, CTAB, CNC inhibit the infection of pepper leaves by *Phytophthora capsici*. CNC@CTAB (3) means that the CTAB density on the CNC surface is 5.50 × 10^−2^ nmol/cm^2^. CNC@CTAB (2) means that the CTAB density on the CNC surface is 3.96 × 10^−2^ nmol/cm^2^. CNC@CTAB (1) means that the CTAB density on the CNC surface is 2.86 × 10^−2^ nmol/cm^2^. CNC is the cellulose nanocrystal. CTAB is the hexadecyltrimethylammonium bromide. CK is the water group.

**Table 1 molecules-24-02467-t001:** Zeta potential and Elemental analysis of the CNC and CNC@CTAB.

Name	Feed Ratio	Zeta Potential	Elemental Analysis	CTAB Surface Density
	(CNC:CTAB)	(mV)	(N%)	(nmol/cm^2^)
CNC	0	−39.4 ± 0.70	0	0
CNC@CTAB (1)	2.50%	−9.5 ± 0.45	0.1 ± 0.03	2.86 × 10^−2^
CNC@CTAB (2)	5.00%	+2.4 ± 0.23	0.14 ± 0.05	3.96 × 10^−2^
CNC@CTAB (3)	10.00%	+7.7 ± 0.32	0.19 ± 0.03	5.50 × 10^−2^

CNC is the cellulose nanocrystal. CNC@CTAB is a material formed by the CTAB and CNC.

**Table 2 molecules-24-02467-t002:** Disease index of the pepper leaves in each treatment group.

Name	Disease Index
CK	16.7 d
CNC	16.7 d
CTAB	13 c
CNC@CTAB (1)	1.5 b
CNC@CTAB (2)	0 a
CNC@CTAB (3)	0 a

CNC is the cellulose nanocrystal. CNC@CTAB is a material formed by the CTAB and CNC. CTAB is the hexadecyltrimethylammonium bromide. CK is the water group. Mean values displayed in each bar followed by different letters are significantly different according to Duncan’s multiple range test (*p* < 0.05).

**Table 3 molecules-24-02467-t003:** Grading criteria for the incidence of pepper leaves.

Level	Degree of Disease
0	No disease
1	The lesions account for less than 10% of the surface area of the leaf
3	The lesions account for 11–30% of the surface area of the leaf
5	The lesions account for 31–50% of the surface area of the leaf
7	The lesions account for 51–75% of the surface area of the leaf
9	The lesions account for more than 75% of the surface area of the leaf

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
