# Peer review of "Cellulose Nanocrystal Surface Cationization: A New Fungicide with High Activity against Phycomycetes capsici"

_molecules, 2019, doi:10.3390/molecules24132467_

Round 1

Reviewer 1 Report

This work describes the surface modification of cellulose nanocrystal with quaternary ammonium salts to produce antifungal systems. Similar systems have been previously reported in literature for antibacterial applications and maybe the novelty of this work is the analysis of the antifungal activity. However, the manuscript needs to be significantly improved before publication.

The manuscript should be carefully revised; there are many typo and grammatical errors, and incomprehensible sentences.

For instance in the introduction section, the authors did not highlight enough the importance of quaternary ammonium salts as fungicide, and sometimes antibacterial and antifungal activity concepts are mixed.  Some parts of introduction section are confusing, and should be rewritten, for instance:

“Quaternary ammonium salts have received extensive attention as a new type of small molecule fungicide [17]. They have advantages of good cell membrane permeability, low toxicity, good environmental stability, long residence time, and strong biological activity [18]. The antibacterial mechanism of these compounds is associated with the positive charge of the quaternary ammonium salt being adsorbed onto the surface of the pathogenic microorganism by electrostatic attraction. This then combines with the negative substance on the cell membrane (such as the phospholipid bilayer) to change the permeability of the cell membrane and cause the leakage of large amounts of the cytoplasm, which eventually leads to cell death [19-20]. However, when quaternary ammonium salts are used as antibacterial agents, they show high instability and lead to drug resistance [21-25]”

The results and discussion section is also not well done. It is difficult to understand. The authors did not explain that the used different percentage of CTAB at the begging and this is only described in Table 1.

The FTIR characterization is very poor and the DXR diffractograms look exactly the same for both samples, even the noise.

There are many errors in the graphs, for instance Figure 1b “intencity”.

The captions of the many figures are incomplete.

Figure S-2 of supporting information is wrong, Figure 2e and Figure 2f are exactly the same.

In summary, the manuscript is not well written and I cannot recommend its publication.

Author Response

Dear reviewer

Thank you for your kindly response. According to the your comments about this manuscript, we have corrected the spelling errors and rewritten the introduction and reanalysed and discussed the results in detail. The specific responses to your comments are depicted as follows:

Point 1: The manuscript should be carefully revised; there are many typo and grammatical errors, and incomprehensible sentences.

Response 1: Thank you for your professional comment. We have carefully rewrite the manuscript to our best, and asked professional English editors to check the English spelling and grammar errors to ensure the quality of the manuscript.

Point 2: For instance in the introduction section, the authors did not highlight enough the importance of quaternary ammonium salts as fungicide, and sometimes antibacterial and antifungal activity concepts are mixed.  Some parts of introduction section are confusing, and should be rewritten.

Response 2: Thank you for your comment. According to your comments, we have rewritten the introduction to make this article easier to understand. we also have highlighted the importance of CTAB as a representative quaternary ammonium antimicrobial agent in resistant microorganisms as following. “Hexadecyl trimethyl ammonium bromide (CTAB), an efficiency anti-microbial surfactant, has been deeply studied and applied for several decades. The quaternary ammonium head of CTAB enable the molecule quickly adsorb onto cell surface. Thus, its hydrophobic moieties could interact with phospholipid bilayer of cell membrane and destroy the self-assemble structure of the pathogeny cell membrane, leading to a disruption of the cell membrane and a release of cell constituents, and that eventually causes to cell death. CTAB has a relative low toxicity and is hard to be resisted by the microorganism. This character allow it to be widely used as an nonspecific anti-microbial agent in water disinfection, bioactive agents, medical apparatus and instruments, antifouling coatings and other antimicrobial works. while, it also been applied on oomycetes and proved had an anti-oomycete activity. However, as an oomycetes inhibitor, its activity is highly depended on its concentration. Therefore, as a kind of pesticides, high application amount of CTAB during plant disease control was still an important problem that remained to be solved.” (Introduction section, second paragraph)

Point 3: The results and discussion section is also not well done. It is difficult to understand. The authors did not explain that the used different percentage of CTAB at the begging and this is only described in Table 1.

Response 3: Thank you for your comment. According to your comments, we had rewritten the result and discussion. We reanalyzed and discussed all the data in this article to show the highlights and innovations of this work. On the other hand, the different percentage of CTAB were used in this work mainly to make the CNC surface with different CTAB density. In the biotest, those CNC@CTAB with different CTAB density had been used to study the effect of surface CTAB density on the anti-oomycete activity of CNC@CTAB. The result of this study is very important for the study of fungicide for agriculture. The manuscript was rewrote as below. (Results and Discussion section)

Point 4: The FTIR characterization is very poor and the DXR diffractograms look exactly the same for both samples, even the noise.

Response 4: Thank you for your comment. According to your comments, we had retested all samples for FTIR and XRD (Figure 1a and 1b) and reanalysed and discussed the date of FTIR and XRD. However, in this work, there was no chemical reaction between CNC and CTAB, the CTAB coated on the CNC surface would not affect the crystal structure of CNC. Therefore, the diffractograms of CNC and CNC@CTAB would not be much different. The manuscript was rewrote as below. “Fourier transform infrared spectroscopy (FTIR) spectroscopy was used to observe CTAB molecules on the particle surface. As it was shown in Figure 1a, after mixing CTAB with CNC, the peaks located at 2900 cm-1 and 2860 cm-1, corresponded to the symmetric and asymmetric C-H stretching vibrations was slightly strengthened, which, compared with the FTIR spectrum of CTAB, was attributed to the formation of CTAB layer at the surface of CNC. Meanwhile, the interaction of CTAB and CNC led to a decrease in the strength of the characteristic peak to 1645 cm-1, corresponding to the decreased water content in the samples, which and was attributed due to the more hydrophobic behavior of CNC@CTAB. This result was also in consistent to the formal reports, and proved that the CTAB was successfully coated on the CNC surface. However, as there was no chemical reaction happened between CNC and CTAB, the coating of CTAB onto CNC surface showed no effect to the crystalline structure of CNC (Figure 1b). The result of X-ray diffraction (XRD)  was also consistent to other related reports. (Results and Discussion section, first paragraph, fourth line to nineteenth line).

Point 5: There are many errors in the graphs, for instance Figure 1b “intencity”.

Response 5: Thank you for your comment. We have corrected the “intencity” into “intensity” and recreated the image. (Results and Discussion section, Figure 1b).

Point 6: The captions of the many figures are incomplete.

Response 6: Thank you for your comment. We have explained the letters and numbers in all the figures of this article. At the same time, we also modified the captions for all the images. The manuscript was rewrote as below. “Figure 2. Comparison of the anti-oomycete activity of CNC@CTAB and CTAB against Phytophthora capsici (a). The inhibition rate statistics of CTAB and CNC@CTAB against Phytophthora capsici (b). Comparison of the anti-oomycete activity of CNC@CTAB with different CTAB density against Phytophthora capsici (c). The inhibition rate statistics of CNC@CTAB with different CTAB density against Phytophthora capsici (d). CNC@CTAB (3) means that the CTAB density on the CNC surface is 5.50×10-2 nmol/cm2. CNC@CTAB (2) means that the CTAB density on the CNC surface is 3.96×10-2 nmol/cm2. CNC@CTAB (1) means that the CTAB density on the CNC surface is 2.86×10-2 nmol/cm2. CTAB is the hexadecyltrimethylammonium bromide. Mean values displayed in each bar followed by different letters are significantly different according to the Duncan's multiple range test (p < 0.05). ** indicated separation among the total CTAB concentration of the CNC@CTAB and pure CTAB at the same concentration by Duncan multiple comparison ( **: p < 0.01). vertical bars indicate standard deviations (n = 3) ± S.E. Figure 3. Curve of the extracellular fluid conductivity of Phytophthora capsici with time after adding CNC, CTAB and CNC@CTAB (with different CTAB density). CNC@CTAB (3) means that the CTAB density on the CNC surface is 5.50×10-2 nmol/cm2. CNC@CTAB (2) means that the CTAB density on the CNC surface is 3.96×10-2 nmol/cm2. CNC@CTAB (1) means that the CTAB density on the CNC surface is 2.86×10-2 nmol/cm2. CNC is the cellulose nanocrystal. CTAB is the hexadecyltrimethylammonium bromide. CK is the water group.  vertical bars indicate standard deviations (n = 3) ± S.E. Figure 4. In vitro test of pepper leaves. CNC@CTAB, CTAB, CNC inhibit the infection of pepper leaves by Phytophthora capsici. CNC@CTAB (3) means that the CTAB density on the CNC surface is 5.50×10-2 nmol/cm2. CNC@CTAB (2) means that the CTAB density on the CNC surface is 3.96×10-2 nmol/cm2. CNC@CTAB (1) means that the CTAB density on the CNC surface is 2.86×10-2 nmol/cm2. CNC is the cellulose nanocrystal. CTAB is the hexadecyltrimethylammonium bromide. CK is the water group. Table2. Disease index of pepper leaves in each treatment group. (Results and Discussion section, The captions of Figure 2, Figure 3, Figure4, and Table 2).

Point 7: Figure S-2 of supporting information is wrong, Figure 2e and Figure 2f are exactly the same.

Response 7: Thank you for your comment. We are very sorry for the mistake. We have corrected the figure. When making Figure S-2, we put two identical pictures together, Figure S-2f should be the width statistical graph of CNC-@CTAB. We had reworked Figure s-2. (Supplementary Material, Figure S-2).

Reviewer 2 Report

In this study, Xiang S examine the role of a bio-based nano-antifungal material (CNC@CTAB) against Phytophthora capsici. However, I am afraid that this study brings little novelty into the field since there is quite a body of literature conserning the development/dynamics/application  of cellulose nanocrystal in the presence of the cationic surfactant Hexadecyl- trimethylammonium Bromide. References:Industrial Crops and products (2015) 65: 45-55; Macromol. Res., (2017) 25: 767-771; Carbohydr Polym (2017) 157:1557-1567; etc. I would like to see this point discuss in great detail by the authors. 

Minor point:

i) the mode of action of CNC@CTAB is in this study little explored;

Author Response

Dear reviewer

Thank you for your kindly response. According to the your comments about this manuscript, we have carefully read the literatures you provided, and compared the literatures with our work in detail. The specific responses to your comments are depicted as follows:

Point 1: I am afraid that this study brings little novelty into the field since there is quite a body of literature concerning the development/dynamics/application  of cellulose nanocrystal in the presence of the cationic surfactant Hexadecyl- trimethylammonium Bromide. References: Industrial Crops and products (2015) 65: 45-55; Macromol. Res., (2017) 25: 767-771; Carbohydr Polym (2017) 157:1557-1567; etc. I would like to see this point discuss in great detail by the authors

Response 1: Thank you for your professional comment. We have carefully read the literatures you provided, and compared the literatures with our work. As is mentioned, quaternary ammonium salts (QA) coated CNC have been fabricated, and studied as a measurement to adjust the surface hydrophilicity of CNC during its application in the enhancement of polymers (Industrial Crops and products (2015) 65: 45-55). And the ability of CTAB layer to increase the monodispersion of CNC was also well studied. (Macromol. Res., (2017) 25: 767-771) However, the anti-microorganism activity of the CNC@CTAB was not discussed in all of these works. Meanwhile, although CTAB was involved in the composite nanoparticle in the last reference (Carbohydr Polym (2017) 157:1557-1567), in this work, the main anti-microorganism agent was silver nanoparticle (or Ag ions it released), but not CTAB. Thus, the problem this work should be faced is the potential safety of silver nanoparticle applied in the textures, while this is also the problem that this particle is hard to be applied in the agriculture production. On the other hand, P. capsici, as a kind of oomycete, is seriously threaten the production of some vegetables, such as pepper. The efficiency of the chemical fungicides is usually limited against the oomycete, and the potential pollution problem of the application of these chemical pesticides is also highly concerned by the researchers. In this work, as CTAB is a kind of traditional anti-microorganism agents, and has been proved has a low toxicity, we believe that by coating CTAB onto CNC surface, the CNC can enrichment CTAB on its surface and interact with cell surface, and thus highly improve the anti-microorganism of the CTAB. Considering the low toxicity of both CNC and CTAB, and their degradability, this particle can be applied as a new kind of efficient green fungicide in the prevention of P. capsici on pepper. We also discussed it in the manuscript as following. “As there are a great amount of negative charges on CNC surface, quaternary ammonium salts (QA) can be coated on CNC surface via electrostatic interaction (Industrial Crops and products (2015) 65: 45-55; Macromol. Res., (2017) 25: 767-771). The composite nanoparticle prepared by CTAB and CNC are proved to reverse the zeta potential of CNC from negative to positive, increase the monodisperse rate of the particle, and also adjust the surface hydrophilicity of the particle. These results can be well applied in the preparation of high-performance CNC-based composite materials (Industrial Crops and products (2015) 65: 45-55; Macromol. Res., (2017) 25: 767-771), but the anti-microbial activity of the CTAB coated CNC (CNC@CTAB) has not been studied, which might be a great useful in the prevention of plant disease. Meanwhile, CTAB can also act as stabilizer of both CNC and inorganic nanoparticles (Carbohydr Polym (2017) 157:1557-1567). For example, CNC/silver nanoparticle/CTAB composite nanoparticle, and is used in PLA film to be applied in anti-bacteria. CTAB in this work can highly improve the stability of silver nanoparticle (SNP) and CNC, while the composite film performed a good anti-bacterial activity (Carbohydr Polym (2017) 157:1557-1567). However, as SNP has been proved to exhibit a high anti-bacterial activity, the ability of CTAB in this material to inhibit microbial is not well studied.” (Introduction section, third paragraph, tenth line to the twenty-third line)

Point 2: The mode of action of CNC@CTAB is in this study little explored.

Response 2: Thank you for your comment. In this work, the CTAB was used to coat on the CNC surface, this strategy successfully turned CTAB, a small molecule anti-oomycete agent, into a bio-based nano-anti-oomycete material (CNC@CTAB). More importantly, by coating CTAB onto CNC surface, the local concentration of CTAB could be increased on CNC surface, while the interaction between cell membrane and the rod-like CNC@CTAB particle would also increase the anti-microbial activity of the CTAB. CNC@CTAB can quickly bind to the oomycete cell membrane and rapidly destroy cell membrane permeability to enhance the anti-oomycete ability of CTAB. However, regarding the mode of action of CNC@CTAB and oomycete cell membranes, we are still in the study.

Reviewer 3 Report

Dear all,

The present study demonstrates the use of cellulose nanocrystal (CNC) as a potential carrier of fungicidal active ingredients, such as hexadecyl trimethyl ammonium bromide (CTAB). The authors thoroughly examined the effective binding of CTAB on CNC, while in vitro and in vivo bioactive assays suggested the effective action of the developed complex (CNC + CTAB) against the plant pathogen Phytophthora capsici. Thus, I suggest the acceptance of the manuscript after minor revisions that I list below:

·      The plant pathogen you use in your bioassays phylogenetically is not a fungus but an oomycete and the species name is Phytophthora capsici and not Phytophthora capsica, so please correct it throughout the text. In addition the species name should be written in italics.

·      Acronyms mentioned in the main text (besides abstract) such as FTIR, XRD, TEM, AFM must be fully written the first time they are mentioned.   

·      The paragraph 4.5 entitled ’Cell permeability measurement’ is poorly written and should be re-constructed in a more comprehensive manner.

·      It is not clearly mentioned or either explained in the manuscript why you mention the percentage of N content in each tested concentration of the “CNC + CTAB” mixture. Why the N content it is so important, since the different concentrations of CTAB used per treatment is written. Please clear this out in the text.

Author Response

Dear reviewer

Thank you for your kindly response. According to the your comments about this manuscript, we have corrected the spelling errors of Phytophthora capsici and rewritten the full name of FTIR, TEM, AFM, XRD. The specific responses to your comments are depicted as follows:

Point 1: The plant pathogen you use in your bioassays phylogenetically is not  a fungus but an oomycete and the species name is Phytophthora capsici and not Phytophthora capsica, so please correct it throughout the text. In addition the species name should be written in italics.

Response 1: Thank you for your comment. We have corrected the “Phytophthora capsica” into “Phytophthora capsici” and corrected the “Pepper” into “Pepper. Also, we have rewritten the manuscript to correct the P. capsici from fungus into oomycete.

Point 2: Acronyms mentioned in the main text (besides abstract) such as FTIR, XRD, TEM, AFM must be fully written the first time they are mentioned.

Response 2: Thank you for your comment. We had added the full name of FTIR, XRD, TEM, and AFM when they first appear in the text.

Point 3: The paragraph 4.5 entitled ’Cell permeability measurement’ is poorly written and should be re-constructed in a more comprehensive manner.

Response 3: Thank you for your comment. We had rewritten the entitled of paragraph 4.5. We have corrected the “Cell permeability measurement.” into “Electrical conductivity assay of cell membrane .” At the same time, we also rewritten the detailed method of testing the cell membrane permeability of Phytophthora capsici. (Materials and Method section, fifth paragraph).

Point 4: It is not clearly mentioned or either explained in the manuscript why you mention the percentage of N content in each tested concentration of the “CNC + CTAB” mixture. Why the N content it is so important, since the different concentrations of CTAB used per treatment is written. Please clear this out in the text.

Response 4: Thank you for your comment. The N content was used to calculate the surface CTAB density of the CNC@CTAB particles. We have changed the N content in the manuscript into surface CTAB density, and clearly described the function of N content in this work as below. “The surface density of CTAB on CNC@CTAB particle was adjusted and calculated by the N content of the CNC@CTAB particle. Then, the inhibition of pure CTAB and CNC@CTAB with different CTAB surface density to P. capsici both in vitro and in vivo were compared.” (Introduction section, fourth paragraph, fourth line to seventh line). “The N% content of CNC@CTAB was used to calculate the surface density of CTAB. Table 1 listed the N% content of CNC and CNC@CTAB with different surface density. It could be found that with the increasing of added CTAB amount, the N% content of the particle was increased, which indicated that the content of CTAB of CNC@CTAB was increased. After calculation, by the controlling of CTAB added amount during the mixing process, the CTAB content of CNC@CTAB could be controlled from 2.86x10-2 nmol/cm2 to 5.50x10-2 nmol/cm2.” (Results and Discussion section, second paragraph, fifth line to tenth line)

Round 2

Reviewer 1 Report

The authors have significantly improved the manuscript, but still there are some errors

For example:

line 68: Discovery in lower case

line 96 This character allows

line 98: while in capital letter

line 183:  remove "dot" (figure 1c-d) it could ( in lower case)

Figure 1a) transmittance  instead of transparency

Author Response

Dear reviewer

Thank you for your kindly response. According to the your comments about this manuscript, we have corrected the errors about the manuscript. The specific responses to your comments are depicted as follows:

Point 1: line 68: Discovery in lower case.

Response 1: Thank you for your professional comment. We have corrected the “Discovery” into “discovery” . (Page 2, line 68)

Point 2: line 96 This character allows

Response 2: Thank you for your professional comment. We have corrected the “This character allow” into “This character allows” . (Page 3, line 96)

Point 3: line 98: while in capital letter

Response 3: Thank you for your professional comment. We have corrected the “while” into “While” . (Page 3, line 98)

Point 4: line 183:  remove "dot" (figure 1c-d) it could ( in lower case)

Response 4: Thank you for your professional comment. We have corrected the “From transmission electron microscope (TEM) images of CNC and CNC@CTAB. (Figure 1c-d) It could also be observed that after coating,” into “From transmission electron microscope (TEM) images of CNC and CNC@CTAB (Figure 1c-d). It could also be observed that after coating,”. (Page 5, line 189)

Point 5: Figure 1a) transmittance  instead of transparency.

Response 5: Thank you for your professional comment. We have corrected the “transparency” into “transmittance” and redrew this picture. (Page 7, Figure 1a, line 218)

Reviewer 2 Report

The authors have improved the manuscript, however I still would like to see more references / discussion related to other published works that used similar systems: e.g. it was already demonstrated that the coating of cationic surfactants/molecules (including CTAB) onto CNC surface affects the zeta potential of the particle.

Author Response

Dear reviewer

Thank you for your kindly response. According to the your comments about this manuscript, we have carefully read the literatures related to our manuscript, and compared the literatures with our work in detail . The specific responses to your comments are depicted as follows:

Point 1: The authors have improved the manuscript, however I still would like to see more references / discussion related to other published works that used similar systems: e.g. it was already demonstrated that the coating of cationic surfactants/molecules (including CTAB) onto CNC surface affects the zeta potential of the particle.

Response 1: Thank you for your professional comment. After receiving your comments, we immediately conducted a literature search and added the reference in the manuscript. In addition, some discussions about quaternary ammonium salt modified CNC were added, including quaternary ammonium salt (CTAB) modified CNC potential, quaternary ammonium salt (CTAB) modified CNC hydrophilicity. We discussed it in the manuscript as following. “On the other hand, the rich positive charge and hydroxyl groups on CNC surface also make the particle could be modified by Quaternary ammonium salt (e.g. CTAB) and applied in the high-performance composite polymers [36-37]. The composite nanoparticles prepared by CTAB and CNC are proved can reverse the zeta potential of CNC from negative to positive, and thus increase the monodisperse rate of the particle, and also adjust the surface hydrophilicity of the particle. These results can be well applied in the adjustment of the compatibility between CNC and polymer matrices to prepare high-performance CNC-based composite materials [36-37]. For example, CTAB can be used to modify the surface of CNC by the electrostatic interaction. After the modification, the zeta potential of CNC is changed from -36.5mV to +6.09mV. Meanwhile, the hydrophobicity of CNC is also increased with the improvement of its dispersibility in PP matrix. the composite CNC can be used to prepare high-performance PP and the Young’s modulus of PP is reported to be increased from 181 MPa to 279 MPa [38]. However, the anti-microbial activity of the CTAB coated CNC (CNC@CTAB) has not been studied, which might be a great useful in the prevention of plant disease.” (Page 3, line 116 to128). “The coating of CTAB or quaternary ammonium salt onto CNC surface also affected the zeta potential of the particle [38,43-45]. As was shown in Table 1, with the increasing of added CTAB amount, the zeta potential of CNC@CTAB was gradually changed from negative (-39.4 mV) to positive (+7.7 mV), this was mainly due to the increase of positive charge amount of CTAB on CNC surface.” (Page 5, line 198 to 202). The location of the added references is in the following two sentences: “Meanwhile, the hydrophobicity of CNC is also increased with the improvement of its dispersibility in PP matrix. the composite CNC can be used to prepare high-performance PP and the Young’s modulus of PP is reported to be increased from 181 MPa to 279 MPa [38]” (Page 3, line 127). “The coating of CTAB or quaternary ammonium salt onto CNC surface also affected the zeta potential of the particle [38,43-45].” (Page 5, line 199).